# Housing and Child Health in South Africa: The Value of Longitudinal Research

**DOI:** 10.3390/ijerph19052497

**Published:** 2022-02-22

**Authors:** Kenneth Chatindiara, Lochner Marais, Jan Cloete

**Affiliations:** Centre for Development Support, University of the Free State, Bloemfontein 9300, South Africa; 2017008844@ufs.ac.za (K.C.); CloeteJS@ufs.ac.za (J.C.)

**Keywords:** housing, child health, longitudinal research

## Abstract

Research investigating the link between housing and health often produces mixed results. It does not always prove that good housing improves health. The results suggest a complex set of factors play a role, and the findings are sometimes contradictory. Two ways of addressing these concerns are longitudinal research, where the relationship between housing and health is measured in the short and medium terms, and a focus on children. We use the children’s housing and health data from the five waves of the National Income and Distribution Survey (NIDS) survey in South Africa, 2008 to 2017. We investigate the effect that continued living in informal housing over the five waves has had on these children’s health. Our results show a statistically significant relationship between prolonged residence in poor housing and poor health outcomes for some health indicators. The results call for a closer understanding of health issues in housing policy in South Africa.

## 1. Introduction

The relationship between housing and health is not always clearly shown by research. We assume that good living conditions are good for health. However, the evidence is mixed at best, with only a few studies pointing to statistically significant relationships. Dependence on cross-sectional surveys and the limited use of longitudinal data mean that many conceptual and methodological gaps remain. Several studies have looked at the relationship between housing and health [1,2,3,4,5,6,7,8]. We argue that longitudinal work is the method most likely to clarify the relationship between housing and health and that more of this kind of work should be performed.

Our study investigates the relationship between the housing conditions in which children grow up and their health later in life. We address the methodological concerns associated with cross-sectional research designs. We compare the housing conditions of children over the five waves of the National Income and Distribution Survey (NIDS) (2008–2017) with their self-reporting of their health in 2017. Although there is increased reference to well-being in the literature, including mental health, we took into account only the physical health indicators.

## 2. The Literature

### 2.1. Defining Inadequate Housing and Health

Mallet et al. define ‘precarious housing’ as housing to which at least two of three conditions apply: Unsuitability, unaffordability and insecurity [9]. ‘Unsuitable housing’ means overcrowding, poor physical conditions and an unsafe environment or poor location. ‘Unaffordable housing’ means the rent or mortgage repayment is too high for the household’s monthly income. ‘Insecure housing’ means insecure tenure and the threat of forced removal. The Constitution of the World Health Organisation (WHO) (1946) defined ‘health’ as ‘a state of complete physical, mental and social well-being and not merely the absence of disease or infirmity’ [10]. This definition has been accepted since then. We deliberately did not include homelessness because homelessness is an under-developed concept in the South African data and literature. The concepts we use in this paper adequately describe the reality on the ground and as captured in the data.

### 2.2. Conceptual Framework

Socioeconomic and political factors affect health and well-being and health equity. Health equity is defined as the absence of unfair and avoidable or remediable differences in health among population groups defined socially, economically, demographically or geographically. Figure 1 shows a conceptual framework for the relationship between housing and health [11]. We have circled the items of particular relevance to this paper. The WHO framework identifies two main determinants of health and well-being: Structural and intermediary. The structural determinants influence intermediary determinants that eventually influence equity in health and well-being. The structural determinants are a person’s socio-political context and socioeconomic position. The intermediary determinants are a person’s material circumstances (including living conditions), behaviour, biological and psychosocial factors and the health system itself. We can thus postulate a link between housing policy (a structural determinant), living conditions (an intermediary determinant) and people’s health and well-being. This paper focuses only on housing and the related living conditions as an intermediary determinant of health and well-being, while acknowledging the broader influences, such as policy, which are beyond our scope here.

### 2.3. Evidence from Research

#### 2.3.1. Housing and Health

Several studies have shown statistical relationships between housing and health [1,3,6,7,13,14,15]. Bad housing has been shown to increase asthma, lead poisoning, injuries and mental health problems [16,17] and acute respiratory infections in children [18]. In a cross-sectional study of housing conditions and child health in Sweden, Oudin et al. found links between dampness and asthma and between mould and headache [19]. In cross-sectional analyses of demographic health surveys across 33 countries, Tusting et al. found that poor housing conditions are likely to cause child mortality since children spend more time at home than adults and are thus more likely to be affected [8].

Socioeconomic factors such as household structure and tenure status have also been shown to affect health. Socioeconomic deprivation and severe household crowding cause psychological distress [4]. Solari and Mare expressed the opinion that the negative effects of being raised in crowded homes can persist throughout life [3]. Howden-Chapman found that homeowners were likely to have better health than people who rent, as tenants are more exposed to housing instability caused by moving from place to place [1]. Using a 2010 postal survey in West Scotland, Ellaway et al. found that housing tenure status (secure housing as opposed to insecure housing) can affect four measures of self-assessed health: Chronic, recent and mental health problems, and health in general. Evaluating the association between the inability to pay a mortgage (an example of unaffordability) and changes in health, Alley et al. found that participants who had fallen behind on their mortgage payments had worse health than those who were up to date [20].

#### 2.3.2. Children, Housing and Health

Children are much more prone to respiratory problems than adults. They spend more time on the floor and put objects in their mouths. They have immature immune systems and metabolisms and fewer opportunities to manage their environment. Early-life exposure to chemical, biological and physical agents comes primarily from the home environment or, as we term it in this paper, unsuitable housing [19].

Longitudinal approaches allow the researcher to study the effect of the continued experience of unsuitable housing at different points in the life course [15]. From a longitudinal analysis of data from the UK National Child Development Survey, Marsh et al. found a significant association between poor physical housing and ill health later in life [15]. Using two composite indices, health and housing deprivation, they showed that housing deprivation (unsuitable housing) leads to poorer health outcomes.

Household characteristics are critical for early childhood development and children’s health, as the home is where children spend a large proportion of their time, and unsuitable conditions can lead to bad health later in life [2,15,17,21].

Concerning the empirical evidence, the literature identifies three examples of unsuitable housing factors that affect children’s health: Household composition, household crowding and the house’s physical condition. Household composition determines future education and socio-emotional behaviour. Leventhal and Newman found that single-parent and blended families have short-term adverse effects on children’s school achievement and health [22] (a ‘blended family’, or ‘stepfamily’, is one in which at least one parent has a child or children who are not related to the spouse or partner biologically or by adoption). Using data from the Botswana Family Health Survey, Ntshebe et al. found a relationship between housing composition, stunting and diarrhea [23]. They found that children not living with both parents were more likely to suffer from stunting and that children living in mother-only households and those with no parents were less likely to have diarrhoea than those living with both parents.

Household crowding is usually measured by the number of occupants divided by the number of rooms. Crowding indicates a socioeconomic disadvantage and creates a stressful home environment [24]. A crowded home affects children, as they have limited space to complete homework or rest, interact with family members, practice skills and develop their own identity [3]. In a study of 10- to 12-year-old children in urban India, Evans et al. found that residential crowding led to high blood pressure in boys and helplessness in girls, but although crowding increased psychological distress, it did not produce serious mental illness [25].

The house’s physical condition plays a role in children’s health. Dockery et al. found that children aged 0 to 9 living in houses had better socio-emotional outcomes than those living in apartments and townhouses [26]. They also found that children living in houses with good external conditions (rated by the interviewer on a scale from 1 to 4: Badly deteriorated, in poor condition, in fair condition, well-kept and in good repair) had better socio-emotional outcomes than those living in houses with poor external conditions. Gifford and Lacombe and Coley et al. also found significant associations between the physical condition of housing and child socio-emotional outcomes [2,27]. Fry found that children who grow up homeless are less likely to perform well in working-memory and decision-making tasks than those who grow up in proper housing [28].

### 2.4. South African Studies

#### 2.4.1. Housing and Health

Several studies show a relationship between housing attributes and health [29,30,31,32]. Marais and Cloete examined the health impacts of the South African housing subsidy programme and found that infrastructure-related factors rather than housing structure affect health outcomes [30]. Nkosi et al., in their analysis of an 11-year panel study undertaken in two suburbs in Johannesburg between 2006 and 2016, found that acute respiratory symptoms were statistically significantly associated with moderately as well as extremely overcrowded houses [32]. Their study further showed that diarrhoea was statistically significantly associated with overcrowding. Shortt and Hammett, in their research into formal and informal housing in the Imizamo Yethu informal settlement in Cape Town, found no statistically significant differences in self-reported physical health but some differences in mental health. In Johannesburg [33], De Wet et al. found, counterintuitively, that people living in informal housing had significantly better health outcomes than those living in formal housing [34]. This was because those living in the informal houses were relatively young and recent migrants.

Crowding is associated with acute respiratory and gastrointestinal symptoms [32]. Using data from the Cape Area Panel Study, Muyeba found that people living in houses constructed by the state’s housing subsidies had better physical health (but, counterintuitively, higher occurrences of teenage pregnancy), implying that health improvement results from better housing quality and a better living environment [35]. The Human Sciences Research Council found that HIV is more prevalent in informal than formal urban areas [36]. The high incidence of disease in informal urban areas can be linked to low socioeconomic status. Gibbs et al. note that the legacy of apartheid and high poverty levels have contributed to poor mental health outcomes amongst black South Africans [37].

#### 2.4.2. Children’s Health

There is very little literature on the effects of housing on children’s health in South Africa. Marais et al., in a study of the relationship between housing conditions and the socio-emotional health of orphans and vulnerable children in South Africa, found a positive relationship between living in informal settlements and better socio-emotional health of these children [38]. They found that crowding was more important than settlement type in determining socio-emotional outcomes. Mathee et al. found that despite the benefits of improved housing and settlement development, large numbers of young children in South Africa still live in hazardous environments that threaten their health [31].

There is no formal system focusing on housing and child health in South Africa. However, the Department of Social work often places orphans and vulnerable children with foster parents receiving a care grant. The focus of this system is to keep children in their communities and to ensure adequate housing and care. This would, for example, apply to cases where both parents have died, and there is a child-headed household. The assumption is that foster care will ensure good housing and health. However, there is no guarantee of this in practice.

## 3. Materials and Methods

The association between housing and health is complex. The causal relationship is difficult to establish, as it can be hidden in or influenced by other covariates. Methodologically, it can be difficult to exclude or control confounding variables [39,40,41].

### 3.1. Background and Data

We used longitudinal data from the National Income Dynamics Study (NIDS), the first national household panel study in South Africa. NIDS is a nationally representative sample of over 28,000 individuals in 7300 households. The Southern Africa Labour and Development Research Unit (SALDRU) at the University of Cape Town implemented the first five waves of NIDS, in 2008, 2010, 2012, 2015 and 2017. We used data from all five waves, available from the DataFirst website (http://www.nids.uct.ac.za/ accessed on 14 September 2021). We used Stata Version 14 (StataCorp, College Station, TX, USA) for analysis.

United Nations Children Fund defines a child as any human being below 18 years. Wave 1 of the NIDS dataset, 2008, included 11,226 children aged 0 to 17 [42]. These respondents were then followed for five waves over nine years. By Wave 5, many of these initial participants were not children anymore (for example, a participant who was 16 in Wave 1 would be 25 by Wave 5). Respondents who dropped out of the survey for various reasons were replaced at each wave. We did not use these replacements in our study. The self-reported data on health was obtained from Wave 5 in 2017. The sample is predominately urban, with approximately 60% of the children in the sample being urban. However, 81% of those in informal housing were urban. South Africa is also 70% urbanised.

The paper focuses only on physical health. The hypothesised drivers of physical health are varied and differ significantly from those of mental health. We chose one set of outcomes upon which to focus the paper.

### 3.2. Defining Adequate Housing for the Study

NIDS required respondents to describe the main house that the household occupies. The answers to this question were then categorised into formal and informal housing. ‘Informal dwelling in the backyard’ and ‘informal house on a separate stand’ were categorised as informal and all other housing types as formal. For our study, we assumed that informal housing represents inadequate housing and formal housing represents adequate (although we acknowledge that this is not always the case). Table 1 shows the numbers living in the two types of housing.

We traced the same 7781 participants over the five waves. Table 2 shows our binary breakdown of inadequate housing and the dummy variables we used in the regression analysis. We considered that living in an informal dwelling for four of the five waves constituted continued experience of poor housing. This was the experience of 669 (8.6%) individuals in the sample.

### 3.3. Defining Health Outcomes for the Study

We used several physical health measures from NIDS as dependent variables, as shown in Table 3. The self-reported answers were binary (yes/no). A value of one was given if the respondent had suffered from a particular symptom in the past 30 days and zero if not. The study used the answers from Wave 5, 2017. The table shows low numbers of reports of these symptoms of poor health except for a fever, body ache, headache and backache.

### 3.4. Statistical Analysis

We used logistic regression for our analysis. Logistic regression establishes the relationship between the dependent variable (health outcomes) and the independent variables (housing characteristics). It was appropriate for our analysis as the dependent variable is binary. It estimates the odds ratios used in interpreting the odds of being in the base category of health compared to being in the target category of health.

We took each health outcome as the dependent variable in the logistic regression. ‘Continued experience of poor housing’ was the main regressor. As covariates in the regression, we used the asset index (the total of 15 household assets), per capita income, gender, age, marital status, and the highest level of education. All models took the form:Health outcome_i_ = β_0_ + β_1_Continued experience of poor housing + β_2_Asset index + β_3_Per capita income + β_4_Gender + β_5_Age + β_6_Marital Status + β_7_Highest level of education + ε

All the βs represent odds ratios, and the health outcomes are those in the 30 days before the survey in Wave 5. The results of this analysis are presented in Table 4 and Table 5.

### 3.5. Limitations

There are four main weaknesses in our study. The first is that the health outcomes are self-reported, which introduces bias, and the second is the construction of the derived housing variables. The dummy variables capture those who experienced the housing condition continuously but do not capture how people transited from one housing condition to another. People might have moved from an informal house to a formal house (positive development), or they might have moved from a formal house to an informal house (negative development). A further weakness is the assumption that formal houses are always associated with better living standards and informal with worse. Thirdly, we use multivariate regression and not a true longitudinal method. Two reasons contributed to this decision. While the sample size appears large at first glance, the number of children in informal housing in any one wave remained small, and between waves there was very little movement of children in or out of informal housing. Further, the number of ill-health events, while varying by symptom, was small overall. Given the very small sample size for some subgroupings and the small observed effect sizes, the use of more complex forms of analysis was not feasible. Fourthly, panel surveys such as NIDS suffer from high sample attrition between survey waves. Attrition results from three cumulative factors: Mortality, migration between waves and survey non-response. In the NIDS data, of the 26,775 sample members interviewed in 2008, 15,673 were re-interviewed in all four subsequent waves, giving an attrition rate for the balanced panel of 41.47%. For this study, the sample members in Wave 1 totaled 11,226 in 2008 and those who were successfully interviewed in Wave 5 totaled 9396 in 2018. For the 9396 sample members successfully interviewed in Wave 1, 6604 were successfully interviewed in Wave 2, 80 refused, 1008 were non-responses at the household level and 8 moved out of South Africa. This gave a Wave 1 to Wave 2 attrition rate of 29.71%. Of the 1096 participants who were successfully interviewed in Wave 1 but fell to attrition in Wave 2, 966 (89.03%) lived in formal dwellings while 119 (10.97%) lived in informal dwellings.

## 4. Results

Here we discuss the results of the logistic regression that were statistically significant at a 90% significance level. Table 4 and Table 5 show the influence of housing on health outcomes 30 days before the survey in Wave 5. The pseudo *R*^2^ was low, ranging from 1.0% (fever model) to 29.3% (arthritis model). Low pseudo *R*^2^ can be interpreted the same way as *R*^2^ (coefficient of determination) in linear regression, representing the variation of the dependent variable explained by the model. Because of the nature and complications of health models, a low pseudo *R*^2^ is expected.

Four health indicators had a statistically significant relationship with the continued experience of poor housing. Participants who experienced this were more likely to have a fever (OR = 1.373, *p* = 0.001), dysuria (OR = 1.852, *p* = 0.045) or swollen ankles (OR = 1.642, *p* = 0.034) than those who did not have this experience. Care should be taken not to draw simplistic conclusions, as these symptoms could relate to poor water and sanitation. On the other hand, participants who had continued experience of poor housing were less likely to have diarrhoea (OR = 0.420, *p* = 0.004) than those who did not have this experience. This finding is contradictory to what was expected.

The asset index was statistically significant in six models. The odds of suffering from a symptom increased as the asset index increased: Persistent cough (OR = 1.879, *p* = 0.033), headache (OR = 2.764, *p* = 0.000), arthritis (OR = 2.151, *p* = 0.039), diarrhoea (OR = 2.329, *p* = 0.068) and swollen ankles (OR = 5.625, *p* = 0.002). The likelihood of suffering from chest pain (OR = 0.447, *p* = 0.028) decreased as the asset index increased. These results thus show no direct relationship between socioeconomic status and health.

Five models showed a statistically significant relationship with per capita income: Headache (OR = 1.000, *p* = 0.010), backache (OR = 1.000, *p* = 0.040), dysuria (OR = 1.000, *p* = 0.059), swollen ankles (OR = 1.000, *p* = 0.028) and weight loss (OR = 1.000, *p* = 0.019). Although the relationship is statistically significant, these results show that the likelihood of suffering from one of these symptoms increased marginally or did not increase at all with an increase in per capita income.

Eight statistically significant models were associated with gender. Women were more likely than men to suffer from fever (OR = 1.284, *p* = 0.000), body ache (1.320, *p* = 0.000), headache (OR = 1.556, *p* = 0.000), backache (OR = 1.610, *p* = 0.000), arthritis (OR = 2.170, *p* = 0.000), swollen ankles (OR = 3.380, *p* = 0.000) or weight loss (OR = 1.331, *p* = 0.052). However, they were less likely to suffer from coughing blood than were their male counterparts (OR = 0.502, *p* = 0.021).

Age was statistically significantly associated with all the health outcome indicators except dysuria. For those that were significant, the likelihood of suffering from any poor health symptom increased as age increased. Previous studies have also found that age plays perhaps the most important role in the health profile of a population [30,31].

In terms of marital status, people living with partners were less likely than married people to suffer from persistent cough (OR = 0795. *p* = 0.000), coughing blood (OR = 0.401, *p* = 0.055) or headache (OR = 0.844, *p* = 0.028). They were, however, more likely to suffer from chest pain (OR = 1.353, *p* = 0.019) or swollen ankles (OR = 1.747, *p* = 0.003). Divorced or separated persons were less likely than married people to suffer from persistent cough (OR = 0.761. *p* = 0.038) and more likely to suffer from backache (OR = 1.343, *p* = 0.011) or swollen ankles (OR = 1.578, *p* = 0.036).

In terms of education, the statistically significant odds ratios were marginal, with all of them being close to one. An increase in education levels marginally decreased the likelihood of suffering from coughing blood (OR = 0.904, *p* = 0.014), chest pain (OR = 0.976, *p* = 0.051), backache (OR = 0.965, *p* = 0.000) or arthritis (OR = 0.953. *p* = 0.000).

## 5. Discussion

The world is urbanising rapidly, and WHO has estimated that the current urban population will double by 2050 [43]. This rapid urbanisation contributes to poor and informal housing. Improvement in housing might improve the quality of life, reduce poverty and improve health [41]. Given that informal houses are associated with poor health, policymakers could look at improvements in housing (inclusive of infrastructure) to deal with public health concerns.

The study’s findings corroborate some international and national literature on the relationship between children’s health and housing. On the one hand, the research results confirm the relationship of poor or unsuitable housing already found in some of the international literature [19]. We strengthen this finding by emphasising the long-term effects. The problem with most housing and health studies is that they only take one data point with respect to housing. By providing a history of poor or unsuitable housing, we look at the consequences over the long term. This is a shortcoming of most of the research in this field. We also confirm findings about the relationship between socio-economic factors, housing and health. Some of the findings point to a relationship between household composition (for example the head of household being divorced or living with a partner and the educational status of the parent or care giver) that were also found in the international literature [22,23,26]. It is the relationship between unsuitable housing and diarrhoea that confirms previous African research [23]. However, we found cases that did not confirm the existing literature. For example, we found no statistical relationships with respect to crowding.

The inequality in housing represented in this paper results from two main causes: The historical exclusion of black people from urban areas and continued problems in providing and servicing informal settlements in South Africa. Under apartheid rule, the government prevented black people from urbanising [44]. When this restrictive policy started to change in the mid-1980s and free movement became possible in the 1990s, it contributed to a large-scale influx of people to informal settlements across South Africa. The historical exclusion of black people from urban areas also meant their exclusion from the urban economy. Therefore, the housing inequalities described in the paper must be seen as part of the reality of South Africa being one of the world’s most unequal countries. Yet there is also an indication that the post-apartheid government has not always embraced urbanisation [45]. Consequently, appropriate urbanisation and informal settlement-upgrade planning have not been adequate.

The WHO states that housing quality has major implications for people’s health, and poor housing conditions are an important determinant of health outcomes [42]. People in our sample who had lived in inadequate housing for prolonged periods (in four or five of the five NIDS waves) had poorer health outcomes for a selection of health indicators. These results align with a study by Kreiger and Higgins that found significant differences in health outcomes between people who had continuous experience of poor housing and those who did not [17]. Our results show that housing can contribute to health inequity in the long term. The value of our study lies in the longitudinal nature of the data, which differs considerably from cross-sectional data. Methodologically, it emphasises the effects of long-term inadequate housing.

Despite the findings discussed above, the main mechanism of poor health is not always clear. The data points to many factors and covariables, other than housing, playing a role. For example, most people living in informal housing do not have ready access to water and sanitation. Earlier work by Marais and Cloete found that the quality of services is the main driver of poor health [30]. Thus, although our study considered inadequate housing, it might indirectly point to insufficient access to services and poor quality of services. Improved infrastructural services (clean water and sanitation) are also important for informal settlement upgrading programmes in South Africa. However, the informal settlement upgrading programme has been slow to take off and does not always follow the programme guidelines [45]. Mainstreaming the informal settlement upgrading programme could help in addressing poor access to infrastructure services.

Overall, there are two ways to improve housing conditions in South Africa: Build new houses for people and upgrade informal settlements. Upgrading might not necessarily lead to improved housing structures but it could improve infrastructure. The South African government has several housing programmes. These have slowed down due to funding constraints. Building new houses has proved an expensive and demanding task for the government and also costly for individuals. Most people living in informal settlements are low-income earners. Besides housing, the government needs to devote attention to providing adequate services. Yet, in addition to these larger policy programmes, support and care programmes can do more to make people aware of the relationship between housing and health. For example, the Department of Social Development has several community outreach programmes to address the role of good housing regarding children’s health. Community-health care workers can perform the same role. For example, healthcare workers can make parents and care workers aware of tuberculosis.

In addition to the potential role of housing structure, a broader framework should be considered. According to WHO, a people-centred health system framework comprises six building blocks: Leadership and governance, health information systems, health financing, health service delivery, human resources and medicines and technology [12]. These building blocks must link to the social policies identified in WHO’s framework shown in Figure 1. In particular, health systems service delivery must be connected to housing as health services must be accessible.

## 6. Conclusions

Our study identified some relationships between housing conditions and health outcomes using a longitudinal approach. This approach contrasts with cross-sectional work, where the research findings are often context specific. Because of collinearity between the three variables designed to measure housing characteristics we originally considered from the literature, only a continued experience of poor housing was used in this study as an independent variable in the logistic regression equation. The results indicate the difficulty of isolating and examining the relationship between housing and health. The relationship does exist, as the models are significant, but the strength of the relationship is weak, as shown by the low *R*^2^. Where the model is significant, the results are mixed.

The paper adds to the literature on the complex relationships between housing and health. It emphasises the importance of longitudinal research and recording the possible health effects of inadequate housing over time. Although this is not the first time this approach has been followed, it is the first time it has been performed in South Africa. However, we should be careful not to think about informal housing only in terms of the housing structure. Informal housing also implies poor access to water and sanitation. Our results call for a closer consideration of housing to address public health concerns

## Figures and Tables

**Figure 1 ijerph-19-02497-f001:**
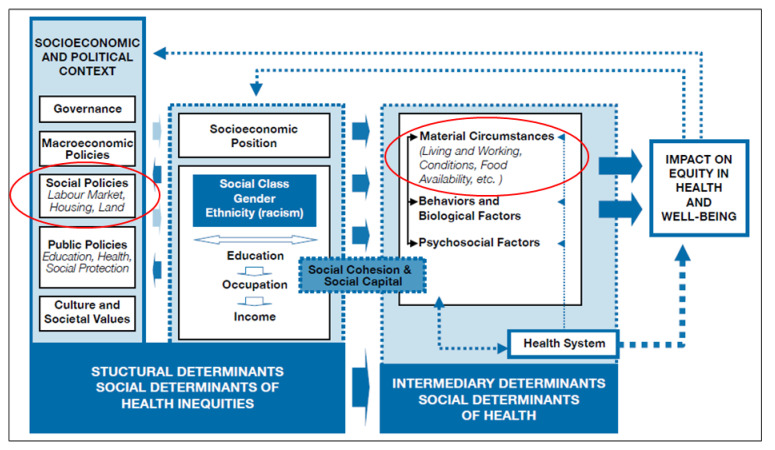
Conceptual framework showing the link between housing and health and well-being [12].

**Table 1 ijerph-19-02497-t001:** Housing conditions and numbers of people aged 0 to 17 in NIDS Wave 1.

Type of Housing	Wave 1 (2008)		Wave 2 (2010)		Wave 3 (2012)		Wave 4 (2015)		Wave 5 (2017)	
	Number	%	Number	%	Number	%	Number	%	Number	%
0 = Informal house	855	7.62	766	7.93	649	6.58	851	8.61	911	9.70
1 = Formal house	10,371	92.38	8888	92.07	9207	93.42	9031	91.39	8485	90.30
Total	11,226	100.00	9654	100.00	9856	100.00	9882	100.00	9396	100.00

**Table 2 ijerph-19-02497-t002:** Housing constructs and variables.

Variable	Description	Number	%
Continued experience of poor housing	0 = If individual lived in a formal dwelling or for less than four waves in an informal dwelling	7112	91.40
	1 = If individual lived in an informal dwelling for four or five waves	669	8.60
	Total	7781	100.00

**Table 3 ijerph-19-02497-t003:** Summary statistics for poor health outcomes.

Health Indicators	1 = Yes		0 = No	
Disease	Number	%	Number	%
Fever	1577	22.62	5395	77.38
Persistent cough	805	11.54	6168	88.46
Coughing blood	58	0.83	6915	99.17
Chest pain	498	7.14	6476	92.86
Body ache	1142	16.39	5826	83.61
Headache	2299	32.96	4677	67.04
Backache	938	13.46	6030	86.54
Arthritis	695	9.97	6273	90.03
Diarrhoea	295	4.23	6678	95.77
Dysuria	86	1.23	6883	98.77
Swollen ankles	257	3.68	6719	96.32
Weight loss	233	3.34	6736	96.66

**Table 4 ijerph-19-02497-t004:** Housing and poor health outcomes in the past 30 days.

Variables and Tests			Fever	Persistent Cough	Coughing Blood	Chest Pain	Body Ache	Head Ache
Model summary	Number of observations		7637	7634	7635	7631	7629	7639
	LR chi squared (14)		86.17	118.05	30.2	210.65	597.33	180.99
	Probability > chi squared		0.000	0.000	0.000	0.000	0.000	0.000
	Pseudo R^2^		0.010	0.022	0.048	0.053	0.088	0.019
Independent variables								
Housing	Continued experience of poor housing	OR	1.373	1.014	0.962	0.914	1.075	0.998
		*p*	0.001	0.919	0.941	0.601	0.544	0.984
Asset index	Asset index	OR	0.824	1.879	0.427	0.447	1.530	2.764
		*p*	0.375	0.033	0.466	0.028	0.105	0.000
Per capita income	Income	OR	1000	1000	1000	1000	1000	1000
		*p*	0.607	0.109	0.163	0.476	0.292	0.010
Gender	Female	OR	1.284	1.096	0.502	1.146	1.320	1.556
		*p*	0.000	0.249	0.021	0.171	0.000	0.000
Age	Age	OR	1.012	1.022	1.021	1.024	1.037	1.012
		*p*	0.000	0.000	0.000	0.000	0.000	0.000
Marital status	Living with partner	OR	0.953	0.795	0.401	1.353	0.904	0.844
		*p*	0.569	0.039	0.055	0.019	0.291	0.028
	Divorced/Separated	OR	0.864	0.761	0.452	1.183	1.178	0.863
		*p*	0.148	0.038	0.108	0.265	0.130	0.111
Education	Education	OR	1.002	0.985	0.904	0.976	0.990	0.998
		*p*	0.779	0.167	0.014	0.051	0.277	0.932
Constant	Constant	OR	0.239	0.061	0.235	0.038	0.041	0.200
		*p*	0.000	0.000	0.000	0.000	0.000	0.000

LR: likelihood ratio; OR: odds ratio.

**Table 5 ijerph-19-02497-t005:** Housing and poor health outcomes in the past 30 days.

Variables and Tests			Back Ache	Arthritis	Diarrhoea	Dysuria	Swollen Ankles	Weight Loss
Model summary	Number of observations		7632	7635	7636	7631	7641	7630
	LR chi squared (14)		724.14	1471.56	28.51	30.66	336.05	45.8
	Probability > chi squared		0.000	0.000	0.000	0.000	0.000	0.000
	Pseudo R^2^		0.120	0.293	0.011	0.031	0.153	0.021
Housing	Continued experience of poor (unsuitable) housing	OR	0.807	1.219	0.420	1.852	1.642	0.801
		*p*	0.134	0.244	0.004	0.045	0.034	0.401
Asset index	Asset index	OR	1.513	2.151	2.329	0.237	5.625	2.325
		*p*	0.150	0.039	0.068	0.113	0.002	0.116
Per capita income	Income	OR	1000	1000	1000	1000	1000	1000
		*p*	0.040	0.569	0.145	0.059	0.028	0.019
Gender	Female	OR	1.610	2.170	1.188	1.462	3.380	1.331
		*p*	0.000	0.000	0.168	0.131	0.000	0.052
Age	Age	OR	1.036	1.072	1.011	1.011	1.043	1.016
		*p*	0.000	0.000	0.019	0.183	0.000	0.002
Marital status	Living with partner	OR	1.250	1.010	0.886	0.759	1.747	1.106
		*p*	0.027	0.938	0.489	0.417	0.003	0.594
	Divorced/Separated	OR	1.343	1.053	0.874	0.916	1.578	0.817
		*p*	0.011	0.702	0.522	0.805	0.036	0.388
Education	Education	OR	0.965	0.953	1.016	0.967	0.994	0.985
		*p*	0.000	0.000	0.382	0.294	0.716	0.429
Constant	Constant	OR	0.025	0.004	0.009	0.029	0.002	0.013
		*p*	0.000	0.000	0.000	0.000	0.000	0.000

LR: likelihood ratio; OR: odds ratio.

## Data Availability

Data can be obtained from: http://www.nids.uct.ac.za/.

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
