# Peer review of "Housing and Child Health in South Africa: The Value of Longitudinal Research"

_ijerph, 2022, doi:10.3390/ijerph19052497_

Round 1
Reviewer 1 Report
The authors indicated the advantage of longitudinal research and its analysis. The manuscript, thus, sheds light on the relationships between health outcomes and the history of housing conditions. In this regard, the health outcomes at the fifth wave were used as dependent variables.
It would be more accurate if the authors could take both income and asset index as dynamic predictors. For example, the level of poverty could be changed across waves.
It would be great if the authors could re-develop Table 4 by the journal's format.
Reviewer 2 Report
Thankyou for the opportunity to review your paper regarding Housing and Child Health in South Africa. Your paper makes an important contribution towards understanding housing as a key determinant of child health. I liked the longitudinal quantitative research design of this research. This shows an innovative and appropriate use of data and evidence. There is a paucity of longitudinal research in this important area of inquiry.
I have some feedback and suggestions and propose that this be addressed prior to publication. The paper introduces some key definitions in sections 2.1 and 2.2. These are important definitions but I was left wondering why the term "homelessness" was left out of your conceptualisation. There are some different international definitions of homelessness, for instance the European Definition of Homelessness and the cultural definition of homelessness (Refer to Chamberlain and Mackenzie), as well as others that could have been considered in your methodology. Much of your paper talks about issues such as inadequate dwellings, the lack of security of the housing, informal dwellings and crowding and over-crowding, which can all contribute towards an individual or family's experience of homelessness. If there is a contextual or methodological reason not to consider homelessness in your analysis, this should be clarified in your article.
Additionally, it would be useful if there was a small paragraph inserted in section 2.4 that provided an overview of the existing housing and child health system in South Africa. This would make your article more accessible to an international audience.
The next piece of feedback is very minor - I noticed full stops that were unnecessary positioned before the subheadings of Results and Discussion. Please remove these.
The Discussion could be strengthened. Your two central recommendations - Building more housing and improving informal housing - are both important recommendations but it would also be interesting to see how policy might be developed to better integrate housing with other health and support services where these are needed. I also felt the discussion could have considered the structural drivers of housing inequality, homelessness and child health. This connection is made earlier in the article but could be re-iterated in the discussion. It is important that we continue to describe the impacts of poverty, racism and inequality on the most marginalised in the community.
I hope this feedback is useful.
Reviewer 3 Report
The authors present an analysis of the association between housing and health among children in South Africa. Using the National Income Dynamics Study, they explore how continuous informal housing impacted several physical health conditions among several thousand children. This topic – and the authors’ findings – are important and timely. However, there are several limitations to the paper as currently written that need to be addressed.
Overall
There is a substantial amount of redundancy in the paper, especially in the introduction. For example, in the first paragraph, the authors state “However, the evidence is mixed at best, with only a few studies pointing to statistically significant relationships.” A few sentences later, they state “…they have found either mixed results, or a positive relationship (good housing creates healthy people), or no relationship.” The authors should review the paper for places that these duplicative statements could be cut.
There are also a few instances of statements that seem contradictory. For instance, in 2.1, the authors state that precarious housing is defined as having two of the three conditions: unsuitability, unaffordability, and insecurity. In 2.3.2, they state that the literature identifies household composition, household crowding, and the house’s physical condition as the three housing factors that impact children’s health. How do these two statements relate to each other? Is precarious housing one of the three factors that impact health? Is it a separate way of thinking about housing and health? The authors should review the paper for places these types of statements could be clarified.
Introduction
At the end of the second paragraph, the authors should give a rationale for why they focus only on physical health outcomes.
2.3.1 The authors state that there are two studies that offer strong evidence to support the connection between housing and health, but do not describe these studies or discuss this further. They also do not explain how the evidence from these studies in stronger than the several studies listed in the next sentence that have shown statistical relationships between housing and health.
Overall, the authors cite a lot of literature in this section, but not all of it is directly relevant to the research question. The authors should consider cutting some to make the Introduction flow better and more succinctly support the point of the study.
Materials & Methods
The authors should check their subsection numbering in this section.
3.2. The authors should clarify the difference between a truly longitudinal study in which multiple measurements on the same individual are included in the model and the analysis they did, in which multiple measurements are combined into a single binary measurement for inclusion in the model. There is still value in this design as they know that the exposure (housing) preceded the outcome (health), but they are not looking at changes over time.
The authors should further justify why they chose this analytic model instead of a truly longitudinal analysis, as well as justifying why they chose living in an informal dwelling for 4 out of the 5 waves (instead of 3 out of 5, or all 5 waves) as their definition of inadequate housing.
In the statistical analysis section (called 3.2 but I think it should be 3.4), they refer to the health outcomes as independent variables in the model. They are the dependent variables.
The authors state in 2.3.2 that one study found the association between housing and health was heavily dependent upon children’s baseline health status. Why was baseline health status not included in the model?
What is the Asset Index?
A third weakness of the study is that children who do not complete all 5 waves are excluded from the analysis. The authors should address was implications this might have for generalizability of results.
Results
The authors should address any concerns about multiple testing and large sample sizes leading to false positive results.
All covariates in the model were associated with more physical health outcomes than inadequate housing was. The authors should either address any implications this has for interpretation of the housing results or spend less text describing these results.
Discussion
The first sentence of the discussion is about urbanization, but this is not set up earlier in the paper. Do all the children included in this analysis live in urban areas? If so, more text should be spent in the Introduction introducing the idea that this paper focuses on housing in an urban setting.
The authors introduce a substantial amount of literature in the Introduction, but only compare their findings to one other paper in the discussion. How do their findings compare to other studies, and what might be the explanation for any differences?
The authors state in the Introduction (and title) that the main value add of this paper is the longitudinal nature of the analysis. This point is reiterated in the Discussion, but should be further expanded upon.
Round 2
Reviewer 1 Report
It is great to see that the authors indicated the study's limitations as a longitudinal approach.
Reviewer 3 Report
No suggestions - accept in present form.